# PAX8 regulon in human ovarian cancer links lineage dependency with epigenetic vulnerability to HDAC inhibitors

Kaixuan Shi[1,2†], Xia Yin[1,3†], Mei-Chun Cai[4†], Ying Yan[5], Chenqiang Jia[2], Pengfei Ma[1], Shengzhe Zhang[2], Zhenfeng Zhang[4], Zhenyu Gu[5], Meiying Zhang[1,3]*, Wen Di[1,3]*, Guanglei Zhuang[1,3]*

[1]State Key Laboratory of Oncogenes and Related Genes, Department of Obstetrics and Gynecology, Ren Ji Hospital, School of Medicine, Shanghai Jiao Tong University, Shanghai, China; [2]School of Biomedical Engineering & Med-X Research Institute, Shanghai Jiao Tong University, Shanghai, China; [3]Shanghai Key Laboratory of Gynecologic Oncology, Ren Ji Hospital, School of Medicine, Shanghai Jiao Tong University, Shanghai, China; [4]State Key Laboratory of Oncogenes and Related Genes, Shanghai Cancer Institute, Ren Ji Hospital, School of Medicine, Shanghai Jiao Tong University, Shanghai, China; [5]GenenDesign Co. Ltd, Shanghai, China

**Abstract** PAX8 is a prototype lineage-survival oncogene in epithelial ovarian cancer. However, neither its underlying pro-tumorigenic mechanisms nor potential therapeutic implications have been adequately elucidated. Here, we identified an ovarian lineage-specific PAX8 regulon using modified cancer outlier profile analysis, in which PAX8-FGF18 axis was responsible for promoting cell migration in an autocrine fashion. An image-based drug screen pinpointed that PAX8 expression was potently inhibited by small-molecules against histone deacetylases (HDACs). Mechanistically, HDAC blockade altered histone H3K27 acetylation occupancies and perturbed the super-enhancer topology associated with PAX8 gene locus, resulting in epigenetic downregulation of PAX8 transcripts and related targets. HDAC antagonists efficaciously suppressed ovarian tumor growth and spreading as single agents, and exerted synergistic effects in combination with standard chemotherapy. These findings provide mechanistic and therapeutic insights for PAX8-addicted ovarian cancer. More generally, our analytic and experimental approach represents an expandable paradigm for identifying and targeting lineage-survival oncogenes in diverse human malignancies.
DOI: https://doi.org/10.7554/eLife.44306.001

*For correspondence:
fudoczhang82@126.com (MZ);
diwen163@163.com (WD);
zhuanglab@163.com (GZ)

†These authors contributed equally to this work

## Introduction

Mammalian development proceeds in a hierarchical manner involving directed differentiation from pluripotent stem cells to lineage-committed precursors, which subsequently propagate and progressively yield terminal progeny that constitute the bulk of functional organs. This process, spatiotemporally co-opting cell fate specification and proliferation, is exquisitely guided by tissue-specific regulators of the gene expression program, oftentimes a remarkably small number of master transcription factors (*Mohn and Schübeler, 2009*). Accumulative evidence suggests that during neoplastic transformation, an analogous dependency may maintain on the altered core regulatory circuitry predetermined by cell of origin where the resultant tumor is derived from *Garraway and Sellers (2006)*. Notable examples of so-called lineage-survival oncogenes include AR (androgen receptor) in prostate adenocarcinoma (*Visakorpi et al., 1995*), CCND1 (cyclin D1) in breast cancer (*Sicinski et al., 1995*), MITF (melanogenesis associated transcription factor) in melanoma (*Garraway et al., 2005*), NKX2-1 (NK2 homeobox 1) in lung adenocarcinoma (*Weir et al., 2007*),

SOX2 (SRY-box 2) in squamous cell carcinomas (*Bass et al., 2009*), ASCL1 (achaete-scute family bHLH transcription factor 1) in pulmonary neuroendocrine tumors (*Augustyn et al., 2014*), OLIG2 (oligodendrocyte transcription factor 2) in malignant glioma (*Ligon et al., 2007*), CDX2 (caudal type homeobox 2) in colorectal cancer (*Salari et al., 2012*), FLT3 (fms related tyrosine kinase 3) in acute myeloid leukemia (*Stirewalt and Radich, 2003*), IRF4 (interferon regulatory factor 4) in multiple myeloma (*Shaffer et al., 2008*), and lately identified PAX8 (paired box 8) in ovarian carcinoma (*Cheung et al., 2011*).

PAX8 belongs to an evolutionarily conserved family of nine nuclear transcription factors (PAX1-PAX9) that mostly play pivotal roles in lineage-dependent regulation during embryogenesis (*Robson et al., 2006*). Mouse genetics studies reveal that PAX8 is restrictedly expressed in developing brain, thyroid, kidney, and Müllerian tract, from which the fallopian tubes, uterus, cervix and the upper third of the vagina originate. As a result, PAX8 knockout models are characterized by hypothyroidism and infertility, due to severe dysgenesis of thyroid and reproductive duct, respectively (*Mansouri et al., 1998*; *Mittag et al., 2007*). Upon completion of ontogenesis, PAX8 expression normally attenuates, but remains detectable in some confined areas throughout adulthood, for example fallopian secretory epithelial cells (*Perets et al., 2013*), possibly to fine-tune tissue homeostasis. Recent evidence presented by Project Achilles supports that PAX8 is a prototype lineage-survival oncogene in epithelial ovarian cancer (EOC), the most lethal form of gynecologic malignancies which is de facto Müllerian, rather than coelomic, in nature based on epidemiological, histopathological, morphological, embryological, molecular, and experimental observations (*Dubeau, 2008*; *Dubeau and Drapkin, 2013*; *Karnezis et al., 2017*). Specifically, PAX8 is frequently upregulated and functionally essential in a major subset of ovarian cancer, regardless of distinct somatic alterations or histologies (*Cheung et al., 2011*). In consequence, there is an emergent interest to exploit PAX8 not only as a diagnostic biomarker but also as a potential therapeutic target across diverse histotypes of EOC. However, both mechanistic underpinnings and pharmacological actionability of PAX8 as an ovarian cancer driver are by far elusive, precluding its clinical translation at the current stage.

In this study, we uncovered a lineage-specific PAX8 regulon in EOC by conducting modified cancer outlier profile analysis (COPA) (*Tomlins et al., 2005*) on RNA sequencing (RNAseq) data of a large cell line panel. The regulatory network was operative, as demonstrated by the PAX8-FGF18 axis in promoting ovarian tumor cell migration. A high-throughput image-based small-molecule screen identified that various histone deacetylase (HDAC) inhibitors, including FDA-approved panobinostat (FARYDAK) and romidepsin (ISTODAX), epigenetically abrogated PAX8 expression and efficaciously suppressed xenografts progression, and therefore, represent promising repurposing opportunities to treat patients affected by epithelial ovarian cancer and potentially other human malignancies driven by lineage-survival oncogenes.

## Results

### COPA identifies a lineage-specific PAX8 regulon in ovarian cancer

Previous studies have applied a powerful bioinformatics method named cancer outlier profile analysis (COPA) on microarray datasets (*Tomlins et al., 2005*; *Tomlins et al., 2008*), and identified novel oncogenic drivers with marked overexpression in just a small fraction of tumor cases. We hypothesized that the emerging transcriptome sequencing profiles would offer unique opportunities to define additional sample-specific dependencies. Therefore, we set out to interrogate a large-scale transcriptional compendium (*Figure 1A*) of ~700 human cancer cell lines (*Klijn et al., 2015*) by performing customized COPA, taking into account the arguably unbiased and accurate nature of RNA-seq-based transcript quantification. Upon fixing relatively stringent thresholds (*Figure 1—figure supplement 1A*), a heterogeneous list of candidate cancer genes were nominated (*Supplementary file 1*), which displayed outlier expression pattern across all cancer cell lines (sample-wise comparison) and among genome-wide mRNA measurements in a given sample (gene-wise comparison) (*Figure 1—figure supplement 1B*). As expected, many established oncogenes in distinct malignancies were detected including ERBB2 in breast cancer, MET in stomach tumor and BCL2 in leukemia/lymphoma. In addition, we found that our analytic framework was exceptionally robust in spotting putative lineage-survival oncogenes. For example, ASCL1, CDX2, IRF4, MITF,

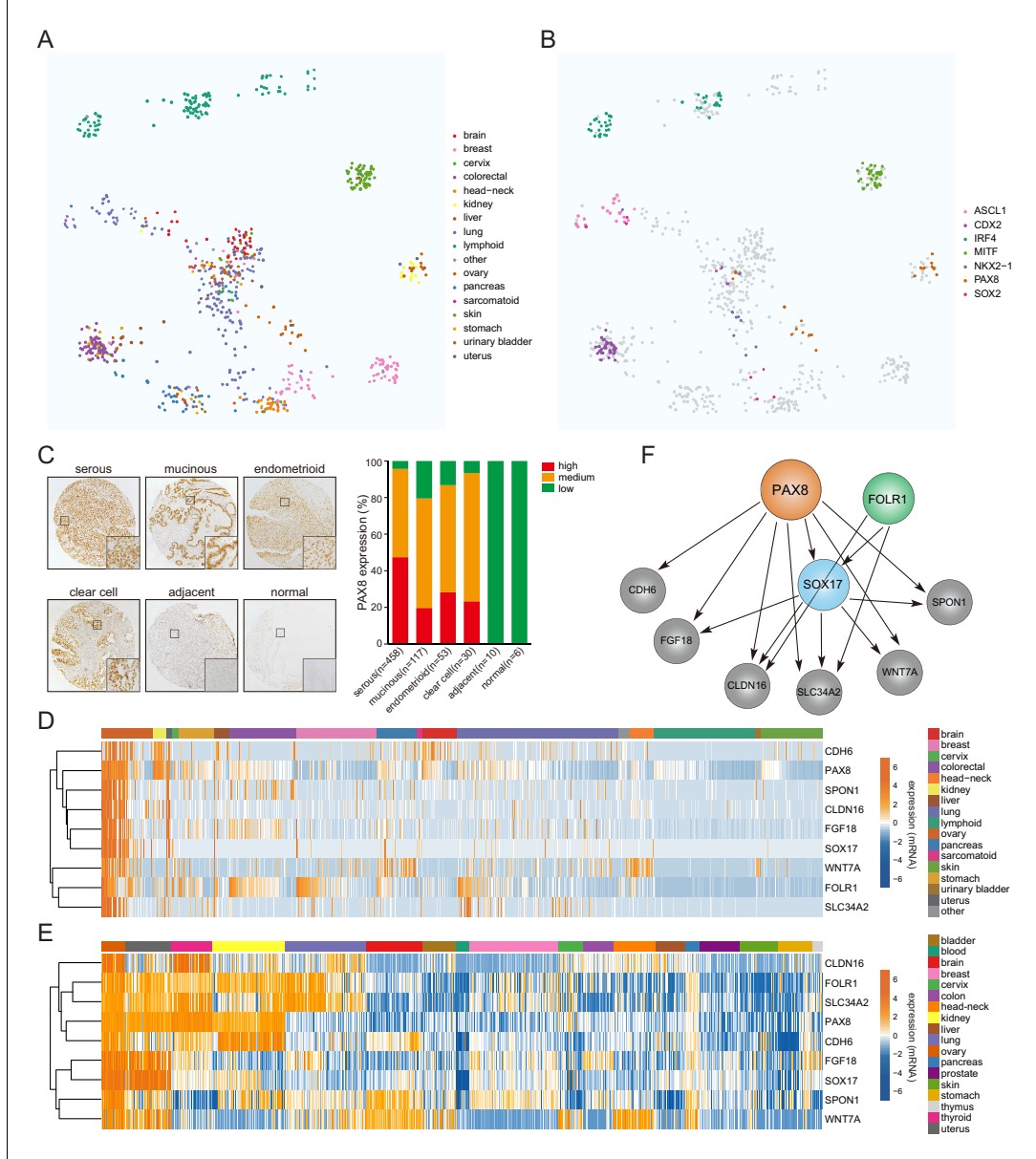

**Figure 1.** COPA identifies a lineage-specific PAX8 regulon in ovarian cancer. (**A**) TumorMap analysis visualizing RNAseq-based transcriptional compendium of human cancer cell lines. Each dot in the map represented a cancer cell line. Colors indicated different tissues of origin. (**B**) The indicated putative lineage-survival oncogenes were overlaid to cancer cell lines in which they were identified as outlier genes. Note that they displayed enrichment in the previously reported tumor types. (**C**) Immunohistochemical staining of PAX8 in tissue microarrays containing 674 ovarian cancer cases. Left panel showed representative IHC images. Quantification in the right panel indicated that PAX8 was expressed in various histotypes of ovarian carcinomas, but not adjacent or normal tissues. (**D**) Confined gene expression of the PAX8 regulon in ovarian cancer cell lines. (**E**) Confined gene expression of the PAX8 regulon in TCGA ovarian tumor tissues. (**F**) Regulatory network of the PAX8 regulon as inferred by individually knocking out each member with CRISPR-Cas9 system and quantifying relative gene expression in ovarian cancer cells.

DOI: https://doi.org/10.7554/eLife.44306.002

The following figure supplements are available for figure 1:

**Figure supplement 1.** COPA identified PAX8 as an outlier gene in ovarian cancer.

DOI: https://doi.org/10.7554/eLife.44306.003

**Figure supplement 2.** PAX8 regulon gene expression in cell lines and TCGA tumor tissues.

DOI: https://doi.org/10.7554/eLife.44306.004

**Figure supplement 3.** PAX8 regulon gene expression in normal tissues and relative expression of each PAX8 regulon upon knocking out the indicated genes.

*Figure 1 continued on next page*

*Figure 1 continued*

DOI: https://doi.org/10.7554/eLife.44306.005

**Figure supplement 4.** Correlation estimation of PAX8 and each regulon gene in ovarian cancer cell lines and TCGA ovarian tumors.

DOI: https://doi.org/10.7554/eLife.44306.006

**Figure supplement 5.** Molecular subgroup analysis and prognosis analysis of PAX8 and PAX8 regulon signature.

DOI: https://doi.org/10.7554/eLife.44306.007

NKX2-1, PAX8 and SOX2 were pinpointed as outliers in cell lines of their corresponding cancer tissue origins (*Figure 1B*).

Building upon the COPA results, we focused on detailedly assessing the lineage-dependency mechanisms in ovarian cancer, a disease of high aggressiveness and few therapeutics (*Coleman et al., 2013*; *Matulonis et al., 2016*). PAX8 represented one of the first described ovarian lineage-specific oncogenes (*Cheung et al., 2011*), and indeed, was focally amplified in a subset of TCGA subjects (*Figure 1—figure supplement 1C*). Consistent with previous reports (*Laury et al., 2011*), immunohistochemical assessment of an EOC tissue microarray (*Figure 1—figure supplement 1D*) containing 674 samples (*Supplementary file 2*) showed that PAX8 was ubiquitously expressed in various histopathological subtypes spanning serous, clear cell, endometrioid and mucinous ovarian carcinomas, but not within the adjacent or normal tissues (*Figure 1C*). In addition to PAX8, COPA complemented with pan-cancer analysis prioritized eight other lineage-restricted outliers (CDH6, CLDN16, FGF18, FOLR1, SLC34A2, SOX17, SPON1, WNT7A) displaying largely confined gene expression in ovarian cancer cell lines (*Figure 1D*; *Figure 1—figure supplement 2A*) and tumor specimens (*Figure 1E*; *Figure 1—figure supplement 2B*). Intriguingly, a query of the normal tissue RNAseq data from Genotype-Tissue Expression (GTEx) project (*GTEx Consortium, 2013*) indicated that similar to PAX8 itself, the 9-gene outlier cluster was enriched in kidney, thyroid and fallopian tubes (*Figure 1—figure supplement 3A*). We subsequently determined their potential hierarchy via individually knocking out each member with CRISPR-Cas9 system and quantifying relative expression of the gene sets in OVTOKO cells (*Figure 1—figure supplement 3B*), followed by validation studies of PAX8 depletion in KURAMOCHI, OV56, COV 362, DOV 13, OAW42 and SKOV3 models (*Figure 1—figure supplement 3C*). These investigations collectively supported that PAX8 resided at the very center of outlier network, transcriptionally activating all other genes but FOLR1, and thus acting as master regulator of a PAX8-centric regulon (*Figure 1F*). In line with this notion, correlation estimation provided evidence that PAX8 was significantly associated with the majority of regulon components at the mRNA level in both EOC cell lines (*Figure 1—figure supplement 4A*) and TCGA primary neoplastic tissues (*Figure 1—figure supplement 4B*). The PAX8 regulon as a signature approximately resembled the differential expression of PAX8 alone in subgroup comparison, that is upregulation in differentiated tumors and downregulation in immunoreactive tumors (*Figure 1—figure supplement 5A*). Neither PAX8 nor PAX8-responsive regulon predicted high-grade serous ovarian cancer patient prognosis using the median expression cutoff (*Figure 1—figure supplement 5B*). Taken together, our comprehensive analyses identified a previously unrecognized lineage-specific PAX8 regulon in epithelial ovarian cancer.

## The PAX8-FGF18 signaling axis promotes tumor cell motility

Although PAX8 had been reported to facilitate ovarian tumorigenesis (*Cheung et al., 2011*; *Elias et al., 2016*; *Ghannam-Shahbari et al., 2018*), the underlying molecular mechanisms remained elusive. We sought to figure out whether the regulon genes played an important role in mediating the endogenous PAX8 function. To this end, we first conducted a microarray experiment on OVTOKO cells transduced with PAX8 small interfering RNA (siRNA) or scrambled control (*Figure 2A*), to systematically probe PAX8 downstream targets and biological effects. Gene set enrichment analysis (GSEA) pinpointed the anticipated suppressed pathways related to cell cycle upon PAX8 knockdown, and of interest, multiple significantly altered signaling modules promoting tumor metastasis (*Figure 2B*). To complement the siRNA-based assays, we genetically knocked out PAX8 in four ovarian cancer cell lines (KURAMOCHI, HEY, SKOV3 and OVTOKO) by employing CRISPR-Cas9 technology with two independent single guide RNA (sgRNA) sequences (*Figure 2C*). In agreement with the microarray data, PAX8 deficiency resulted in not only impaired cell proliferation (*Figure 2D*) but also defective transwell migration (*Figure 2E*). To exclude the possible cross-

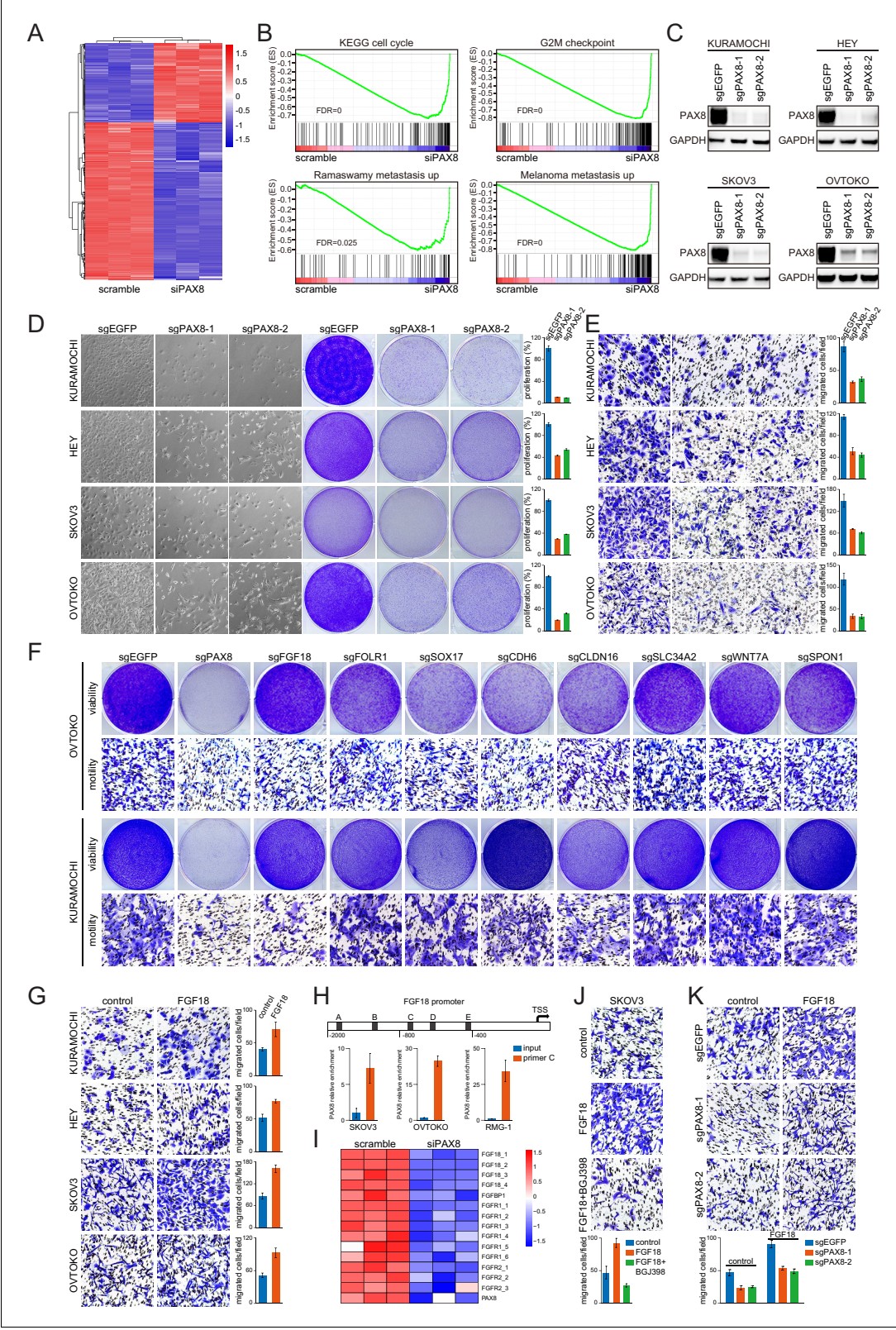

**Figure 2.** The PAX8-FGF18 signaling axis promotes tumor cell motility. (A) Hierarchical clustering of the microarray data in OVTOKO cells transduced with PAX8 siRNA or scrambled control. (B) GSEA plots indicated downregulation of cell cycle or tumor metastasis related gene sets upon PAX8 knockdown in OVTOKO cells. (C) PAX8 was knocked out in KURAMOCHI, HEY, SKOV3 and OVTOKO cells using CRISPR-Cas9 system with two independent sgRNAs, and the indicated proteins were analyzed by immunoblotting. (D) PAX8 was knocked out in KURAMOCHI, HEY, SKOV3 and

*Figure 2 continued on next page*

*Figure 2 continued*

OVTOKO cells. Cell growth was assayed by phase-contrast microscopy or crystal violet staining. The bar graphs showed quantification of crystal violet staining. Each column represented the mean value of three biological replicates, and error bars indicated standard deviation. (E) Transwell migrated cells as stained by crystal violet. The bar graphs showed quantification of cell migration. Each column represented the mean value of three biological replicates, and error bars indicated standard deviation. (F) Cell viability and motility upon individual knockout of indicated PAX8 regulon genes. (G) Transwell cell migration with or without FGF18 stimulation (100 ng/ml). The bar graphs showed quantification of cell migration. Each column represented the mean value of three biological replicates, and error bars indicated standard deviation. (H) ChIP-qPCR using primer set C illustrated PAX8 binding at FGF18 promoter. (I) FGF18 or FGFR gene expression in microarray analysis of OVTOKO cells transduced with PAX8 siRNA or scrambled control. (J) FGF18-stimulated transwell cell migration with or without FGFR inhibitor BGJ398 (1 μM). The bar graphs showed quantification of cell migration. Each column represented the mean value of three biological replicates, and error bars indicated standard deviation. (K) FGF18-stimulated transwell cell migration with or without PAX8 depletion. The bar graphs showed quantification of cell migration. Each column represented the mean value of three biological replicates, and error bars indicated standard deviation.

DOI: https://doi.org/10.7554/eLife.44306.008

The following figure supplements are available for figure 2:

**Figure supplement 1.** PAX8 deficient cells showed impaired cell motility and enhanced focal adhesions.

DOI: https://doi.org/10.7554/eLife.44306.009

**Figure supplement 2.** PAX8 knockdown decreased FGF18 production and cellular motility.

DOI: https://doi.org/10.7554/eLife.44306.010

reaction between cellular viability and motility, live imaging of tumor cells was performed and confirmed the quiescent manifestation in the absence of PAX8 gene (*Figure 2—figure supplement 1A*). Accordingly, immunofluorescent staining of paxillin and F-actin demonstrated abnormally enhanced focal adhesions in PAX8-depleted cancer cells (*Figure 2—figure supplement 1B*). These findings implicated PAX8 in stimulating both cell division and cell migration.

We then performed a phenogenotypic screen using OVTOKO and KURAMOCHI to explore whether manipulating PAX8 regulon genes would impact on ovarian tumor cell viability or motility (*Figure 2F*). Notably, PAX8, SOX17 and CLDN16 each appeared to sustain cell proliferation, albeit to varied extents, whereas PAX8, FGF18 and CDH6 evidently contributed to cell migration in these two models. Of particular importance, FGF18 was specifically involved in regulating ovarian cancer cell migration without affecting tumor outgrowth, which was verifiable across different models (*Figure 2G*; *Figure 2—figure supplement 2A*). Chromatin immunoprecipitation coupled to detection by quantitative real-time PCR (ChIP-qPCR) proved that PAX8 directly bound to the FGF18 promoter (*Figure 2H*) and drove FGF18/FGFR gene expression (*Figure 2I*). Moreover, FGFR inhibitor BGJ398 abrogated FGF18-induced cell migration (*Figure 2J*). These results raised the possibility that FGF18, as a novel bona fide effector in the PAX8 regulon, was responsible for its pro-migratory action. Backing up this hypothesis, exogenous supplement of FGF18 protein partly rescued the cell migration defects caused by sgRNA-mediated PAX8 depletion (*Figure 2K*). The functional outcome of PAX8 knockdown on FGF18 production and cellular motility was completely recapitulated in siRNA-based studies (*Figure 2—figure supplement 2B*). We concluded that PAX8 promoted ovarian cancer cell migration, at least partially, through activating the autocrine FGF18-FGFR signaling pathway.

## Class I HDAC inhibition antagonizes PAX8 expression

While our data strongly argued that PAX8 was a rational target in EOC to inhibit both tumor growth and dissemination, the pharmaceutical development of therapeutic strategies against transcription factors had been extremely challenging. The observation that PAX8 could be distinctly and exclusively detected in nuclei by immunofluorescence (*Figure 3—figure supplement 1A*) prompted us to devise an image-based approach for proposing drug candidates which might antagonize its activity. We initially surveyed PAX8 levels in a diverse array of cell lines (*Figure 3A*), and chose to use KURAMOCHI as the model system because it expressed abundant PAX8 and represented archetypal high-grade serous carcinoma (*Domcke et al., 2013*), the most prevalent and lethal histotype of ovarian cancer (*Matulonis et al., 2016*). KURAMOCHI was subjected to a small-molecule screen with a library of 180 FDA-approved or clinically relevant compounds (*Supplementary file 3*), followed by high-throughput imaging of PAX8 and DAPI staining (*Figure 3B*). The top 15 ranked agents to decrease PAX8/DAPI intensity ratios were selected for subsequent analysis (*Figure 3C*). Among

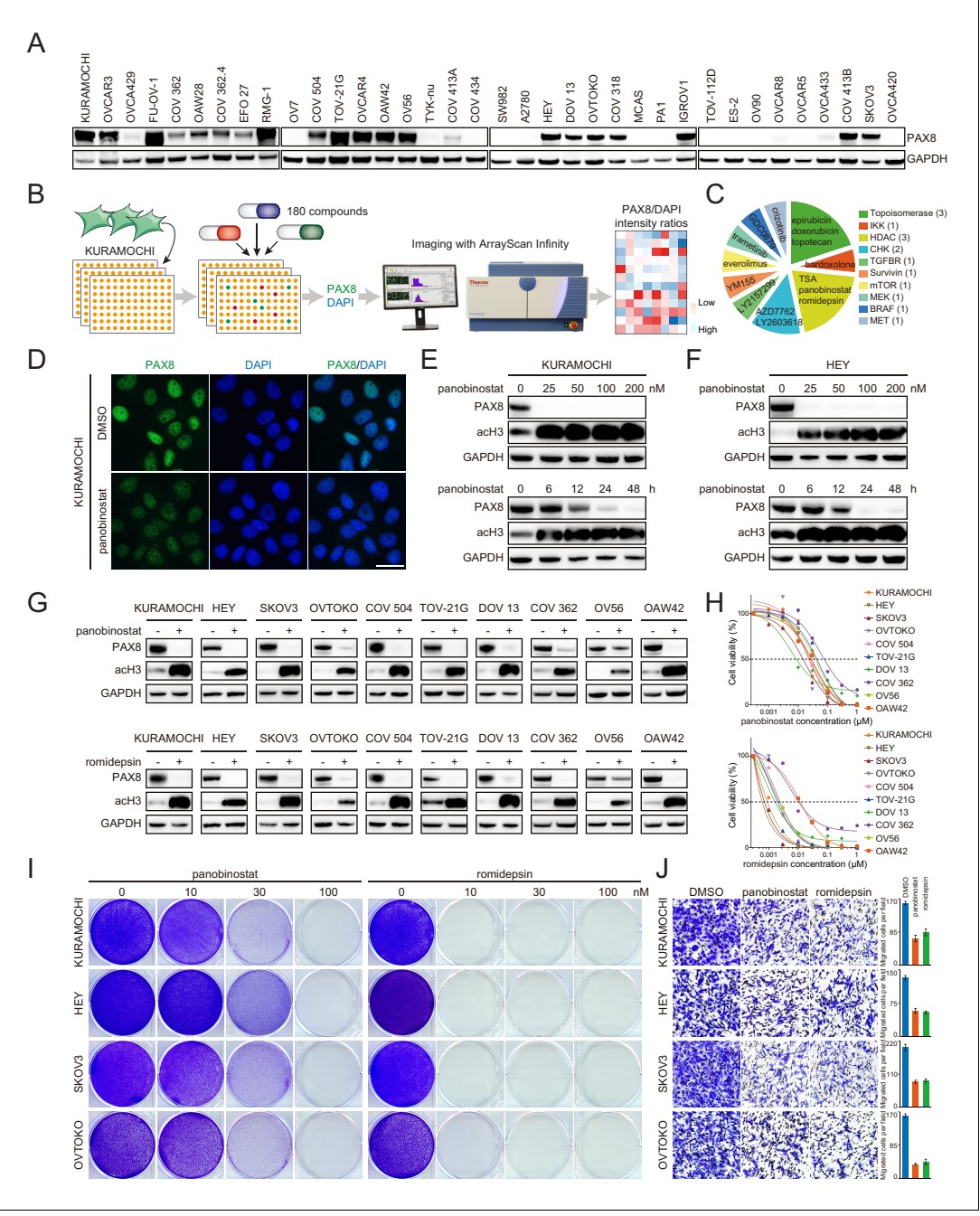

**Figure 3.** Class I HDAC inhibition antagonizes PAX8 expression. (**A**) PAX8 immunoblotting across a panel of ovarian cancer cell lines. (**B**) Schematic overview of high-throughput image-based small-molecule screen in KURAMOCHI cells. (**C**) Top 15 ranked compounds reducing PAX8/DAPI intensity ratios and their corresponding targets. (**D**) Immunofluorescent images of PAX8 staining in the presence or absence of panobinostat treatment (100 nM). (**E**) KURAMOCHI cells were treated with panobinostat and cell lysates were analyzed by immunoblotting. Panobinostat reduced PAX8 protein levels in dose- (treatment duration: 48 hr) and time- (drug concentration: 100 nM) dependent manners. (**F**) HEY cells were treated with panobinostat and cell lysates were analyzed by immunoblotting. Panobinostat reduced PAX8 protein levels in dose- (treatment duration: 48 hr) and time- (drug concentration: 100 nM) dependent manners. (**G**) Indicated ovarian cancer cells were treated with panobinostat (100 nM) or romidepsin (50 nM) for 48 hr, and cell lysates were analyzed by immunoblotting. (**H**) Cell viability in the indicated panel of ovarian cancer cell lines treated with various concentrations of panobinostat or romidepsin for 72 hr. (**I**) KURAMOCHI, HEY, SKOV3 and OVTOKO cells were treated with panobinostat or romidepsin, and were analyzed by crystal violet staining. (**J**) Transwell cell migration with or without HDAC inhibitors (panobinostat: 100

*Figure 3 continued on next page*

*Figure 3 continued*

nM; romidepsin: 50 nM). The bar graphs showed quantification of cell migration. Each column represented the mean value of three biological replicates, and error bars indicated standard deviation.

DOI: https://doi.org/10.7554/eLife.44306.011

The following figure supplements are available for figure 3:

**Figure supplement 1.** Class I ,but not class II, HDAC inhibitors reduced PAX8 expression.

DOI: https://doi.org/10.7554/eLife.44306.012

**Figure supplement 2.** The effects of HDAC gene knockout and PAX8 overexpression.

DOI: https://doi.org/10.7554/eLife.44306.013

them, the most prominent and appealing drug class was HDAC inhibitors, multiple of which earned FDA approval for the treatment of lymphoma and myeloma (*Jones et al., 2016*; *West and Johnstone, 2014*). As an example, panobinostat potently diminished PAX8 signal within 24 hr in KURAMOCHI (*Figure 3D*), HEY, SKOV3 and OVTOKO cells (*Figure 3—figure supplement 1B*). Western blotting ensured that panobinostat administration reduced PAX8 protein at dose- and time-dependent manners in KURAMOCHI (*Figure 3E*) and HEY (*Figure 3F*) cells. Since three classical HDAC (HDAC1-11) classes existed and panobinostat was a pan-HDAC inhibitor, we further discovered that class I (romidepsin and entinostat), but not class II (tubacin and TMP195), HDAC blockade specifically eliminated PAX8 accumulation (*Figure 3—figure supplement 1C*). Corroboratively, PAX8 was downregulated upon simultaneous knockout of three key class I HDAC members (HDAC1-3) (*Figure 3—figure supplement 2A*), which led to impaired cell viability and motility (*Figure 3—figure supplement 2B*). Extending our findings out of the screen, we evaluated the effects of panobinostat (pan-HDAC inhibitors) and romidepsin (class I HDAC inhibitors) in a larger panel of ovarian cancer cell lines. Either of the treatments efficiently eradicated PAX8 protein (*Figure 3G*) as well as PAX8-positive tumor cells (*Figure 3H*). Besides verified growth inhibition by crystal violet staining (*Figure 3I*), cell migration was likewise sensitive to panobinostat or romidepsin exposure (*Figure 3J*). Importantly, PAX8 overexpression via lentiviral transduction (*Figure 3—figure supplement 2C*) partially reversed the antagonistic effects of HDAC inhibitors (*Figure 3—figure supplement 2D*). Hence, both pharmacologic and genetic evidence supported a role of class I HDAC in regulating PAX8 expression and function.

## HDAC inhibitors epigenetically disrupt PAX8 transcription

We integrated chromatin immunoprecipitation followed by next-generation sequencing (ChIPseq) and RNA sequencing (RNAseq) technologies to understand how HDAC antagonists imposed rigorous constrains on PAX8 expression. Given that class I HDAC proteins (mainly HDAC1-3) were best known to possess lysine deacetylase activity, and that enhancer domains marked by acetylated histone were reported to play vital roles in the control of gene transcription (*Creyghton et al., 2010*; *Whyte et al., 2013*), we first performed ChIPseq in KURAMOCHI, SKOV3, and COV 413B cells using an antibody recognizing histone H3K27 acetylation (H3K27ac). Strikingly, H3K27ac occupancy profiles unveiled a typical super-enhancer structure around PAX8 promoter, which displayed exceptionally intensive signal compared to its gene body (*Figure 4A*). HDAC inhibitors probably perturbed the enhancer topology, as suggested by aberrantly altered H3K27ac distribution ratios within PAX8 promoter zone relative to the intron regions (*Figure 4B*; *Figure 4—figure supplement 1A*). In accordance with previous implications of super-enhancers in governing cell-type-specific genes (*Whyte et al., 2013*), panobinostat or romidepsin treatment resulted in rapid and pronounced downregulation of PAX8 transcripts (*Figure 4C*). At the transcriptome-wide scale, a time-course RNAseq assay in KURAMOCHI cells revealed that PAX8 was one of the hypersensitive genes to HDAC inhibitors, but not actinomycin D (*Figure 4—figure supplement 1B*). Both HDAC inhibition and PAX8 depletion caused a global change of gene expression (*Figure 4—figure supplement 1C*), including the above-mentioned PAX8 regulon (*Figure 4D*). Although HDAC inhibitors exhibited comparably larger effects than PAX8 sgRNAs (*Figure 4E*), GSEA pinpointed that the most differentially expressed genes were associated with generally overlapping pathways across various conditions (*Figure 4F*). We defined a PAX8 signature by merging uniformly deregulated targets upon PAX8 inactivation (*Supplementary file 4*), and subsequently showed that this gene set was

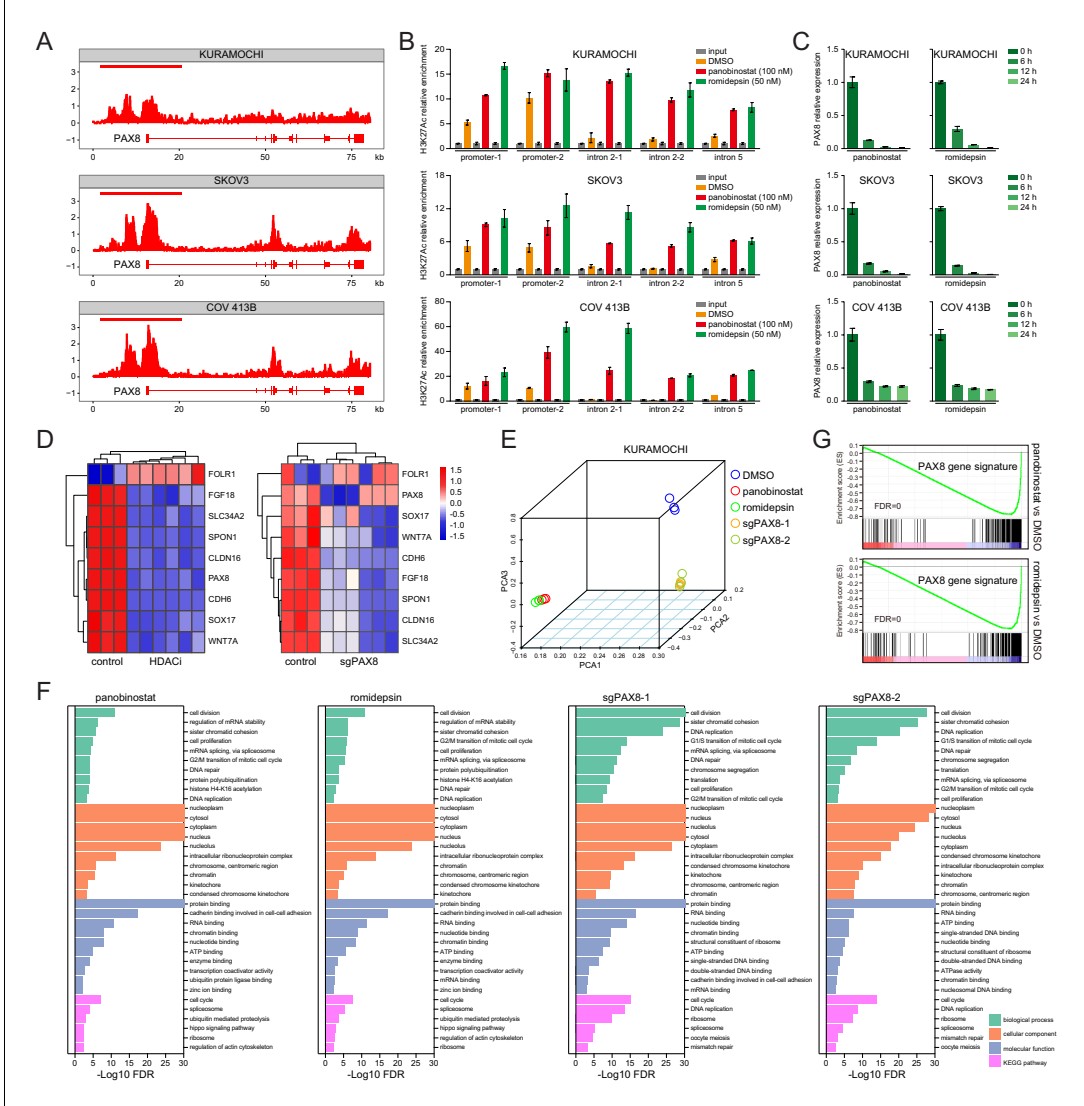

**Figure 4.** HDAC inhibitors epigenetically disrupt PAX8 transcription. (**A**) ChIPseq profiles for H3K27ac occupancy of PAX8 gene locus in KURAMOCHI, SKOV3 and COV 413B cells. The x-axis showed gene tracks, and the y-axis showed the signal of H3K27ac binding within 50 bp bins in units of reads per million bin (rpm/bin). (**B**) ChIP-qPCR quantification of H3K27ac relative enrichment in PAX8 promoter or intron regions as compared to input signals. Each column represented the mean value of three biological replicates, and error bars indicated standard deviation. (**C**) Quantitative PCR analysis of PAX8 gene expression in KURAMOCHI, SKOV3 and COV 413B cells treated with panobinostat (100 nM) or romidepsin (50 nM). Each column represented the mean value of three biological replicates, and error bars indicated standard deviation. (**D**) Heatmaps of PAX8 regulon gene expression in KURAMOCHI cells as measured by RNAseq. (**E**) Principal component analysis (PCA) of RNAseq data in KURAMOCHI cells with HDAC treatment or PAX8 depletion. (**F**) Gene ontology categories and KEGG pathways overrepresented in differentially expressed transcripts that were inhibited upon HDAC treatment or PAX8 depletion in KURAMOCHI cells. (**G**) GSEA plots indicated downregulation of PAX8 gene signature upon HDAC treatment in KURAMOCHI cells.

DOI: https://doi.org/10.7554/eLife.44306.014

The following figure supplements are available for figure 4:

**Figure supplement 1.** HDAC inhibitors altered H3K27ac distribution and resulted in rapid downregulation of PAX8.

DOI: https://doi.org/10.7554/eLife.44306.015

**Figure supplement 2.** HDAC inhibitors suppressed lineage survival oncogenes expression and cell proliferation in lung cancer.

DOI: https://doi.org/10.7554/eLife.44306.016

preferentially enriched in HDAC-modulated transcripts (*Figure 4G*), reinforcing the link between HDAC and PAX8. We concluded that through reshaping epigenetic markers, HDAC inhibitors disrupted PAX8 transcription and its downstream gene expression.

Having identified the HDAC-mediated epigenetic regulation of lineage-dependent PAX8 in ovarian cancer, we became aware that an analogous concept might be broadly applicable to other tumor types. Here, we considered two additional scenarios, namely NKX2-1 in lung adenocarcinoma and SOX2 in squamous cell carcinomas. Gratifyingly, quantitative PCR indeed uncovered a notable reduction of NKX2-1 or SOX2 levels along with HDAC blockage in all tested lung adenocarcinoma or squamous cell carcinoma models, respectively (*Figure 4—figure supplement 2A*). As anticipated, these cell lines responded well to panobinostat or romidepsin regimens (*Figure 4—figure supplement 2B*). We speculated that many lineage-survival oncogenes were likely vulnerable to epigenetic therapies.

## Targeting PAX8 or HDAC shows antitumor effects in mice

To test the in vivo function of PAX8 regulon, we generated intraperitoneal xenotransplant models using firefly luciferase-labeled HEY or SKOV3 cells in which PAX8 was genetically depleted with two independent sgRNAs. Reminiscent of our in vitro findings, tumor development in mice was

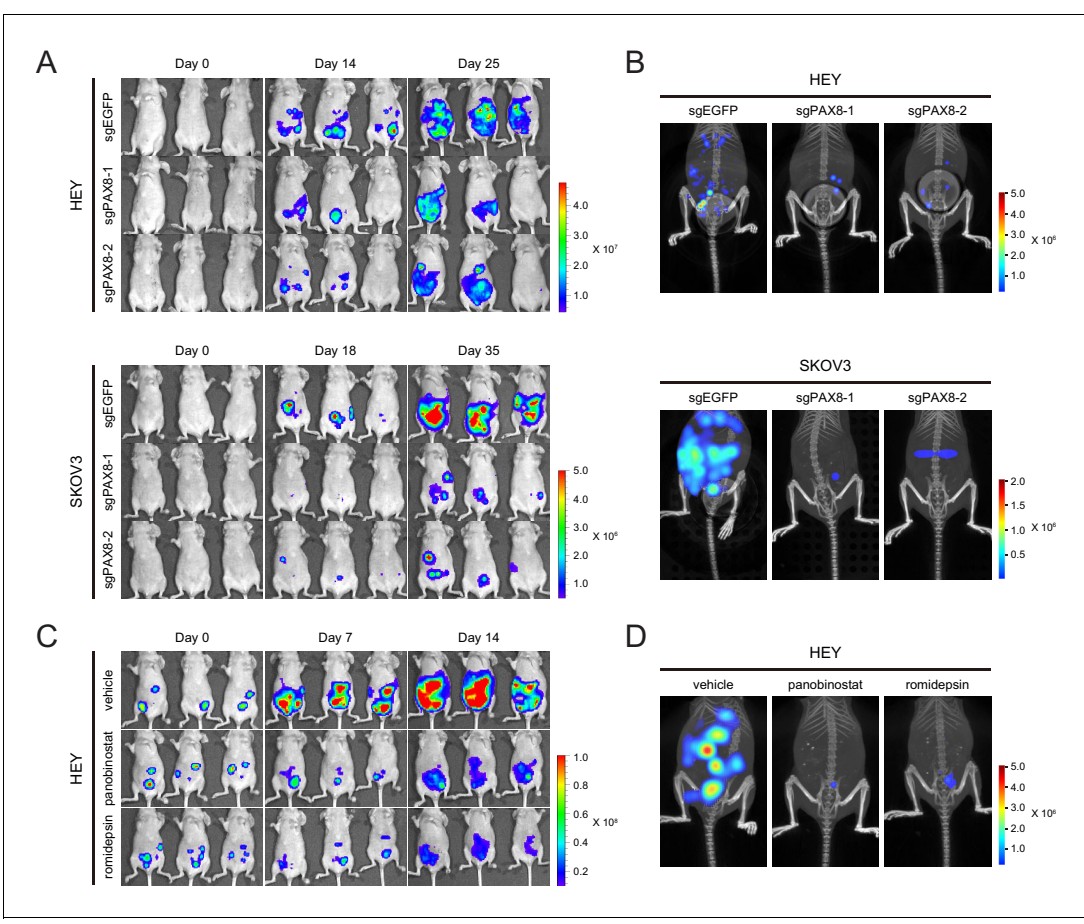

**Figure 5.** Targeting PAX8 or HDAC shows anti-tumor effects in mice. (**A**) HEY or SKOV3 cells with or without PAX8 depletion were labeled with firefly luciferase and implanted intraperitoneally. Tumor development in mice was monitored by bioluminescence imaging. (**B**) Three-dimensional reconstruction of tumor lesions in mice illustrated impaired abdominopelvic spreading upon PAX8 depletion. (**C**) HEY cells labeled with firefly luciferase were implanted intraperitoneally and exposed to panobinostat or romidepsin treatment. Tumor development in mice was monitored by bioluminescence imaging. (**D**) Three-dimensional reconstruction of tumor lesions in mice illustrated impaired abdominopelvic spreading upon HDAC inhibition.

DOI: https://doi.org/10.7554/eLife.44306.017

significantly hampered upon PAX8 knockout compared with the control group (*Figure 5A*), as measured by the bioluminescence imaging. Moreover, three-dimensional reconstruction of neoplastic lesions illustrated evidently impaired abdominopelvic spreading (*Figure 5B*), thus confirming the pro-migratory activity of PAX8. Similarly, when the HEY xenografts were treated with panobinostat or romidepsin, both tumor burden (*Figure 5C*) and metastatic dissemination (*Figure 5D*) were markedly reduced by the two HDAC inhibitors. Taken together, targeting PAX8 or HDAC allowed for efficacious interventions of ovarian cancer growth and invasiveness in mice.

## HDAC inhibitors and chemotherapy synergistically induce tumor death

Platinum-based chemotherapy remains the mainstay of clinical management for epithelial ovarian cancer. From a translational standpoint, we sought to determine whether HDAC inhibition could improve the therapeutic index of standard care. To this end, we applied suboptimal dosage of cisplatin and HDAC inhibitors in KURAMOCHI, HEY, SKOV3 and OVTOKO cells. Remarkably, addition of panobinostat or romidepsin to cisplatin agonistically repressed PAX8 expression and elicited enhanced cleavage of poly-(ADP-ribose) polymerase (PARP), a marker of cells undergoing apoptosis, accompanied by profoundly downregulated pro-mitotic cyclin D1 (*Figure 6A*). Interestingly, we also observed proteolytically cleaved gasdermin E (GSDME), a newly recognized executor of cell pyroptosis (*Lu et al., 2018*; *Rogers et al., 2017*; *Wang et al., 2017*), and consistently, lactate dehydrogenase (LDH) release was detected, indicating plasma membrane rupture and leakage (*Figure 6—figure supplement 1A*). Therefore, combinatorial chemotherapy and HDAC inhibitors induced concurrent apoptotic and pyroptotic cell death in ovarian cancer. As a result, cell viability was synergistically inhibited by polytherapies, as demonstrated by dose matrix experiments (*Figure 6B*) and crystal violet staining (*Figure 6C*). We further generated a chemo-resistant SKOV3 clone and found that both PAX8 expression (*Figure 6—figure supplement 1B*) and cell proliferation (*Figure 6—figure supplement 1C*) remained susceptible to panobinostat or romidepsin exposure.

To validate our observations, we assembled a patient-derived xenograft (PDX) cohort containing 221 models from seven different types of solid cancers, and RNAseq indicated that PAX8 was restrictedly expressed in two PDXs of epithelial ovarian cancer (*Figure 6—figure supplement 1D*). The two ovarian PDX models (EOC-002 and EOC-004) were established and each randomized into four study arms, receiving vehicle control, cisplatin, panobinostat, and combo therapy, respectively. Following ~4 weeks of treatment, the cisplatin +panobinostat combination imparted a more significant suppression of tumor progression compared to the single agents, with cancer regression achieved in EOC-002 (*Figure 6D*) and stable disease observed in EOC-004 (*Figure 6E*). Immunohistochemical staining revealed reduced PAX8 expression in the presence of panobinostat, which was associated with deficient cell proliferation and survival (*Figure 6F*). On the basis of these data, we proposed that chemotherapy combined with HDAC inhibitors might represent a valuable therapeutic option to produce robust and durable benefit for ovarian cancer patients featured by PAX8 dependency.

## Discussion

This study has extended previous knowledge of PAX8 as a diagnostic biomarker and lineage-survival oncogene in at least two dimensions. First, in the sense of tumor biology, we delineate a functional lineage-specific PAX8 regulon in ovarian carcinoma, as exemplified by the PAX8-FGF18 regulatory axis acting autocrinely to facilitate cell migration. Second, from a therapeutic perspective, we provide mechanistic rationale and experimental evidence for targeting PAX8-mediated lineage-dependency with epigenetic therapies, such as pan- or class I HDAC inhibitors. These findings have immediate translational implications for repurposing FDA-approved HDAC antagonists to treat PAX8-driven epithelial ovarian cancer, and in addition, open up new avenues for prospective basic and clinical investigations.

By implementing deliberate computational and epistatic analyses, a nine-gene PAX8 regulon as cancer outlier has been described and revealed to exhibit lineage-restricted expression pattern in both neoplastic ovarian tissues and normal fallopian tubes. Conceivably, these PAX8 regulon elements are involved in decisive events that determine cell fate and phenotype during Müllerian duct organogenesis, and in the context of oncogenesis, may undergo dysregulation to assist ovarian malignant transformation and tumor progression. Indeed, most regulon genes besides PAX8 have

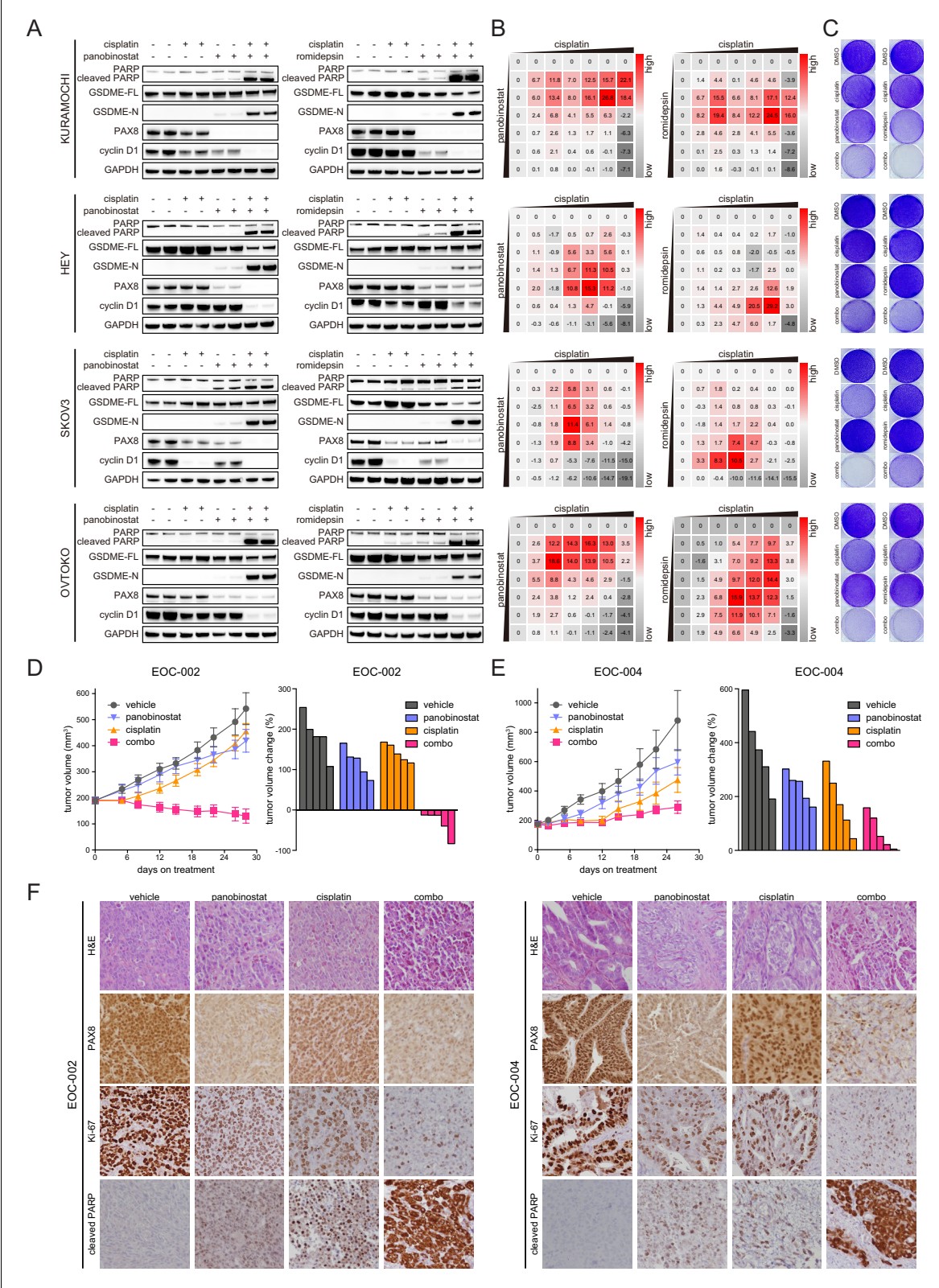

**Figure 6.** HDAC inhibitors and chemotherapy synergistically induce tumor death. (A) KURAMOCHI, HEY, SKOV3 and OVTOKO cells were treated with cisplatin (2 μM) and HDAC inhibitors (panobinostat: 50 nM; romidepsin: 25 nM) as indicated, and cell lysates were analyzed by immunoblotting. (B) Heatmaps of bliss synergy scores demonstrated synergistic activities of cisplatin and HDAC inhibitors in KURAMOCHI (cisplatin: 0, 0.25, 0.5, 1, 2, 4, 8 μM; panobinostat: 0, 12.5, 25, 50, 100, 200, 400 nM; romidepsin: 0, 0.5, 1, 2, 4, 8, 16 nM), HEY (cisplatin: 0, 0.1, 0.3, 1, 3, 10, 30 μM; panobinostat: 0, 1, 3,

*Figure 6 continued on next page*

*Figure 6 continued*

10, 30, 100, 300 nM; romidepsin: 0, 0.0625, 0.125, 0.25, 0.5, 1, 2 nM), SKOV3 (cisplatin: 0, 0.1, 0.3, 1, 3, 10, 30 µM; panobinostat: 0, 1, 3, 10, 30, 100, 300 nM; romidepsin: 0, 0.0625, 0.125, 0.25, 0.5, 1, 2 nM) and OVTOKO (cisplatin: 0, 0.25, 0.5, 1, 2, 4, 8 µM; panobinostat: 0, 12.5, 25, 50, 100, 200, 400 nM; romidepsin: 0, 0.5, 1, 2, 4, 8, 16 nM). (C) KURAMOCHI, HEY, SKOV3 and OVTOKO cells were treated with cisplatin (2 µM) and HDAC inhibitors (panobinostat: 50 nM; romidepsin: 25 nM) as indicated, and were analyzed by crystal violet staining. (D) Tumor growth curves were shown for the PDX EOC-002 model treated with indicated regimens. The right panel indicated relative tumor volume changes at the end point versus the treatment start point. (E) Tumor growth curves were shown for the PDX EOC-004 model treated with indicated regimens. The right panel indicated relative tumor volume changes at the end point versus the treatment start point. (F) Representative images of hematoxylin and eosin (H&E) and immunohistochemistry staining for PAX8, Ki-67 or cleaved PARP.

DOI: https://doi.org/10.7554/eLife.44306.018

The following figure supplement is available for figure 6:

**Figure supplement 1.** HDAC inhibitors induced cell pyroptosis and restrained chemo-resistant ovarian cancer cell proliferation.
DOI: https://doi.org/10.7554/eLife.44306.019

been individually characterized to play a key part in EOC. For example, FGF18 and WNT7A were proposed to promote ovarian tumor aggressiveness (*Wei et al., 2013*; *Yoshioka et al., 2012*); FOLR1 and CDH6 were recently highlighted as antigens to design antibody-drug conjugates against ovarian cancer (*Bialucha et al., 2017*; *Moore et al., 2017*); CLDN16 and SLC34A2 represented ovarian cancer-specific transcripts (HOSTs) (*Rangel et al., 2003*); SOX17 and SPON1 were ambiguously tied to ovarian carcinomas (*Bapat et al., 2010*; *Pyle-Chenault et al., 2005*). Surprisingly, the plausible direct interaction of these components with PAX8 has not been explored in detail, nor have the biological consequences of such connections. Here, we only begin to address the functional impact of PAX8-FGF18 on cell motility, but envision that the illuminated PAX8 regulon may affect one or another hallmark of ovarian cancer. Of note, the entire PAX8-responsive network in EOC has unlikely been exhaustively captured, and with the collaborative efforts of Human Cancer Models Initiative to generate new cell lines, a more thorough dissection of expansive PAX8 molecular program will be feasible and unmask the potential for future drug discovery.

Although nearly all EOC patients respond well to first-line platinum-based chemotherapy, relapse is inevitable and recurrent disease is usually recalcitrant to other available remedies. By far, searching for novel actionable alterations and matched therapeutics has gained limited success in ovarian cancer (*Yap et al., 2009*). Based on the central role of PAX8 in EOC, we employed fluorescence imaging of cell nuclei in a high-throughput format and identified HDAC antagonists to vigorously block PAX8 expression. Importantly, we inferred the turbulent H3K27ac-marked enhancers as the persuasive mechanism of action underpinning the above observations, in contrast with the canonical view regarding HDAC proteins as transcriptional repressors owing to their histone deacetylase activity. This conjecture is in line with the seminal work demonstrating that super-enhancers activate crucial cell identity genes and are sensitive to perturbation (*Whyte et al., 2013*), and also suggests that other lineage-survival oncogenes may likewise be susceptible to epigenetic interventions, a notion supported by our preliminary tests in lung cancer. Numerous HDAC small molecules are being developed or have entered clinical use for the management of hematopoietic malignancies, and therefore, could be in principle rapidly translated to control lineage-associated carcinogenesis. As a prelude, we showed dramatic single-agent efficacy of panobinostat or romidepsin opposing ovarian xenografts, and more encouragingly, panobinostat in conjunction with cisplatin exerted synergistic antitumor effects in cell culture and PDX models. Considering that current HDAC inhibitors suffer from long-term safety concerns, poor pharmacokinetic profiles, and mixed results in earlier ovarian cancer trials (*Jones et al., 2016*; *Khabele, 2014*; *West and Johnstone, 2014*), these results point to an exciting opportunity to combine them with chemotherapy and meanwhile look upon PAX8 regulon as promising biomarkers for improved treatment outcomes in systematic or intraperitoneal settings.

In conclusion, taking PAX8 as an example, we present the proof-of-concept research that integrates bioinformatics-assisted, screen-guided and mechanism-driven approaches to tackle the critical challenge of understanding and targeting lineage-survival oncogenes in often difficult-to-treat neoplasms. Our initial identification of an operative PAX8 regulon in ovarian carcinoma links lineage dependency with epigenetic vulnerability to HDAC inhibition. This study may serve as a template for interrogating lineage-addictions on specific traits including transcription factors in other tumor types,

and should inform therapeutic development and drug repositioning to eventually deliver clinical benefits for cancer patients.

# Materials and methods

## Key resources table

| Reagent type (species) or resource | Designation | Source or reference | Identifiers | Additional information |
|---|---|---|---|---|
| Antibody | anti-PAX8 | GeneTex | GTX101583 | |
| Antibody | anti-PARP | Cell Signaling Technology | cat#9532 | |
| Antibody | anti-cleaved PARP | Cell Signaling Technology | cat#5625 | |
| Antibody | anti-GAPDH | Cell Signaling Technology | cat#8884 | |
| Antibody | anti-Paxillin | Abcam | ab32084 | |
| Antibody | anti-HDAC1 | Santa Cruz | sc-81598 | |
| Antibody | anti-HDAC2 | Abcam | ab32117 | |
| Antibody | anti-HDAC3 | Cell Signaling Technology | cat#3949 | |
| Antibody | anti-H3K27Ac | Abcam | ab4729 | |
| Antibody | anti-GSDME | Abcam | ab215191 | |
| Antibody | anti-cyclin D1 | Abcam | ab134175 | |
| Commercial assay or kit | Alexa Fluor 594 phalloidin | ThermoFisher Scientific | A12381 | |
| Peptide, recombinant protein | Recombinant human FGF18 | Biovision | cat#4082–25 | |
| Commercial assay or kit | FGF18 ELISA kit | Lifespan | LS-F23007 | |
| Commercial assay or kit | CytoTox 96 Non-Radioactve Cytotoxicity Assay Kit | Promega | G1780 | |
| Commercial assay or kit | D-luciferin | Promega | P1042 | in PBS |
| Chemical compound, drug | Panobinostat | Selleck Chemicals | S1030 | in DMSO |
| Chemical compound, drug | Romidepsin | Selleck Chemicals | S3020 | in DMSO |
| Chemical compound, drug | Entinostat | Selleck Chemicals | S1053 | in DMSO |
| Chemical compound, drug | TMP195 | Selleck Chemicals | S8502 | in DMSO |
| Chemical compound, drug | BGJ398 | Selleck Chemicals | S2183 | in DMSO |
| Cell line (H. sapiens) | HEK293T | ATCC | | |
| Cell line (H. sapiens) | HEY | ATCC | | |
| Cell line (H. sapiens) | SKOV3 | ATCC | HTB-77 | |
| Cell line (H. sapiens) | KURAMOCHI | JCRB | JCRB0098 | |

*Continued on next page*

*Continued*

| Reagent type (species) or resource | Designation | Source or reference | Identifiers | Additional information |
|---|---|---|---|---|
| Cell line (*H. sapiens*) | OVTOKO | JCRB | JCRB1048 | |
| Recombinant DNA reagent | LentiCRISPRv2 | PMID:25075903 | | |
| Recombinant DNA reagent | pLenti7.3/V5-DEST Gateway Vector | ThermoFisher Scientific | V53406 | |

## Cell culture

The HEK293T and ovarian cancer cell lines were originally obtained from American Type Culture Collection (ATCC) or Japanese Collection of Research Bioresources Cell Bank (JCRB), where cell characterization was authenticated using polymorphic short tandem repeat (STR) profiling. No mycoplasma contamination was found. Luciferase-labeled stable cell lines were generated by infecting HEY or SKOV3 cells with lentiviral construct expressing firefly luciferase, followed by puromycin selection (5 µg/mL) for one week. Cells were maintained in RPIM1640 (Life Technologies) supplemented with 10% fetal bovine serum (Gibco). All cells were cultured at 37°C in a saturated humidity atmosphere containing 5% $CO_2$.

## Antibodies and reagents

The following antibodies were used: anti-PAX8 (GeneTex, GTX101583), anti-PARP (#9532, Cell Signaling), anti-cleaved PARP (#5625, Cell Signaling), anti-GAPDH (#8884, Cell Signaling), anti-Paxillin (ab32084, Abcam), anti-HDAC1 (sc-81598, Santa Cruz), anti-HDAC2 (ab32117, Abcam), anti-HDAC3 (#3949, Cell Signaling), anti-H3K27Ac (ab4729, Abcam), anti-GSDME (ab215191, Abcam), anti-cyclin D1 (ab134175, Abcam). Alexa Fluor 594 phalloidin (A12381) was from ThermoFisher Scientific. Recombinant human FGF18 (#4082–25) was from Biovision. FGF18 ELISA kit (LS-F23007) was from Lifespan. CytoTox 96 Non-Radioactve Cytotoxicity Assay Kit (G1780) was from Promega. In vivo grade D-luciferin (P1042) was purchased from Promega. All inhibitors were bought from Selleck Chemicals. For in vitro assays, inhibitors were reconstituted in DMSO (Sigma-Aldrich) at a stock concentration of 10 mM.

## Microarray and quantitative PCR analysis

Total RNA was isolated with Trizol reagent (Invitrogen) following the manufacturer's protocol (Invitrogen). The Affymetrix human genome U133 Plus 2.0 microarray was used for gene expression profiles. Probe-level data were background-corrected, normalized and summarized using the robust multi-array average method. Differential gene expression was performed with linear models for microarray data (Limma) implemented in BioConductor. For quantitative PCR, 1 µg of total RNA was reverse-transcribed to cDNA and subjected to RT-PCR using a master-mix with SYBR-green (Invitrogen) on the Applied Biosystems ViiA7 machine. Relative expression levels of each gene were normalized to GAPDH as the endogenous control for all experiments. At least three biological replicates were performed for each condition. The related primers were shown in *Supplementary file 5*.

## Western blot

Cells were lysed in cold RIPA buffer (50 mM Tris-HCl pH 7.4, 150 mM NaCl, 1% NP-40, 0.1% SDS, 1 mM EDTA, proteinase inhibitors and phosphatase inhibitors). The cell lysates were quantified using a BCA Protein Assay Kit (Thermo Scientific) and subjected to SDS-PAGE. Separated proteins were transferred to nitrocellulose membranes and immunoblotted with primary antibodies at 4°C overnight. Membranes were then incubated for 2 hr with horseradish peroxidase conjugated secondary antibody (Cell Signaling) and visualized by chemiluminescence with ChemiDoc XRS+ system (Bio-Rad).

## Tissue microarray

Four ovarian cancer tissue microarray slides were purchased from US Biomax Inc. The tissue slides were dewaxed with xylene and rehydrated through graded alcohol. Heat-epitope antigen retrieval was performed in citric sodium (pH 6.0) for 30 min in a steam pressure cooker, followed by cooling down to room temperature. The endogenous peroxidase activity was blocked with 3% $H_2O_2$ in methanol for 10 min. The slides were blocked with goat serum and then incubated with the primary antibody against PAX8 (1:200) at 4°C overnight, followed by incubation with biotinylated secondary antibody for 1 hr at room temperature. Antigen visualization was performed with 3,3'-diaminobenzi-dine (DAB) chromogen.

## Immunofluorescence microscopy

Cells were fixed for 15 min in 4% paraformaldehyde and permeabilized with 0.1% Triton X-100 (in PBS) for 10 min. After three PBS washes, cells were blocked with 2% BSA for 30 min at room temperature, and incubated with primary antibodies diluted in 2% BSA at 4°C overnight. After three PBS washes, the cells were incubated with secondary antibodies (Invitrogen) in the dark for 1 hr. Cells were washed three times with PBS in the dark, stained with DAPI (Invitrogen) and mounted in Prolong Gold Antifade Reagent (Invitrogen). The immunofluorescent staining was observed using a confocal microscope (Leica).

## High-throughput inhibitor screen

Cells were seeded at optimal density in 96-well plates and treated with the indicated inhibitors for 48 hr at the same concentration (500 nM). Medium was discarded and cells were fixed in 4% paraformaldehyde for 15 min and permeabilized with 0.1% Triton X-100 for 10 min. After PBS washes for three times, cells were blocked with 2% BSA at room temperature and incubated with anti-PAX8 (1:600 dilution in 2% BSA) at 4°C overnight. After three PBS washes, cells were incubated with Alexa Fluor 488-conjugated secondary antibodies at room temperature in dark for 1 hr. Cells were then stained with DAPI and washed with PBS three times. PAX8/DAPI intensity ratio was calculated for each well using ArrayScan Infinity (Thermo Scientifc) according to the manufacturer's instructions.

## Transwell cell migration assays

Cells were starved in serum free medium overnight. Transwell chambers with 8 µm pore membranes (Corning) were placed in 24-well culture plates. Approximately $0.4–1 \times 10^5$ cells were suspended in 200 µL serum free medium, seeded in the upper chambers with 600 µL RPMI1640 medium supplemented with 10% FBS (or 2% FBS and 100 ng/mL rhFGF18 in the FGF18 stimulation assays) in the bottom wells, and incubated at 37°C for 10–24 hr. The migrated cells attached to the lower membranes of the chambers were fixed with 4% paraformaldehyde and stained with crystal violet. The top membranes were wiped clean with cotton swabs. The number of the migrated cells was counted under the microscope (200X).

## Cancer outlier profile analysis and TumorMap

The cancer outlier profile analysis was performed as described (*Tomlins et al., 2005*) with modifications for RPKM (reads per kilobase per million mapped reads) values of the RNA sequencing data. Briefly, the analytic pipeline was as follows: (1) the RPKM values were scaled by the median expression of a given gene across samples. The transformed values were termed rowFold and set the cutoff of >32; (2) the RPKM values were scaled by the median expression of all genes in a sample. The transformed values were termed colFold and set the cutoff of >64; (3) genes that exceeded both rowFold and colFold thresholds were defined as cancer outlier genes. For TumorMap analysis, RNA sequencing data were submitted to the UCSC TumorMap website (https://tumormap.ucsc.edu/). XY coordinates of TumorMap points were downloaded and ggplot2 package in R was employed to redraw the graph.

## RNA sequencing

KURAMOCHI cells were genetically edited with sgRNAs targeting PAX8, or treated with panobino-stat (100 nM) or romidepsin (50 nM) for 24 hr. Total RNA was isolated with RNeasy plus mini kit reagents following the manufacturer's instructions (Qiagen). RNA purity and integrity were assessed

by the NanoPhotometer spectrophotometer (Implen) and the RNA 6000 Nano Assay Kit of Bioanalyzer 2100 system (Agilent Technologies), respectively. A total amount of 3 μg RNA was used for the RNA library preparations for each sample. Sequencing libraries were generated using the rRNA-depleted RNA with NEBNext Ultra Directional RNA Library Prep Kit for Illumina (NEB, USA). The index-coded libraries were clustered on a cBot Cluster Generation System using TruSeq PE Cluster Kit v3-cBot-HS (Illumina) and sequenced on an Illumina Hiseq X Ten platform to generate 150 bp paired-end reads (Novogene, Beijing). Clean data were obtained by removing low quality reads and reads containing adapters or ploy-N sequences. All the downstream analyses were based on the clean data with high quality. Index of the reference genome was built using Bowtie v2.2.3 and paired-end clean reads were aligned to the reference genome using TopHat v2.0.9. HTSeq (v0.6.1) was used to count the reads mapped to each gene. And then FPKM of each gene was calculated based on the length of the gene and reads count mapped to this gene. Differential expression analysis was performed using DESeq2 R package (1.18.1). Transcripts with adjusted p-values of <0.05 were assigned as differentially expressed genes.

## Chromatin immunoprecipitation and sequencing

Cells were crosslinked with serum-free medium containing 1% formaldehyde for 10 min at room temperature. Crosslinking was stopped by adding 1/20 vol of 2.5 M glycine. The cells were harvested using silicon scraper, and washed by ice cold PBS. Pellets were suspended and rocked in three lysis buffers sequentially as previously described (*Lee et al., 2006*). The lysates were sonicated using an ultrasonic processor VC505 (Sonics and Materials) to get fragments ranging from 200 to 500 bp in length. Then 50 μL lysates were taken as whole cell extract (WCE). The remaining chromatin fragments were incubated with antibody-coated magnetic beads overnight. To prepare beads, 100 μL of magnetic beads (Millipore) were incubated with 10 μg PAX8 or H3K27Ac antibodies overnight. The fragments-attached beads were rinsed several times, and incubated in elution buffer (50 mM Tris-HCl pH 8.0, 10 mM EDTA, 1% SDS) at 65°C for 15 min. Both the elution samples and corresponding WCE samples were then incubated overnight at 65°C. Samples were sequentially treated with RNase A and Proteinase K, and then extracted with QIAquick PCR Purification Kit according to the manufacturer's instructions (Qiagen). Before sequencing, immunoprecipitated DNA samples were used to confirm the enrichment of target DNA fragments by means of quantitative real-time PCR. The enrichment of target sequences in ChIP material was calculated relative to the GAPDH-negative control, and normalized to their relative amplification in WCE DNA. The samples were subjected to library preparation and sequenced on an Illumina Hiseq 2000 platform, with 20 million 50 bp reads generated (Novogene, Beijing). Index of the reference genome was built and clean reads were aligned to the reference genome using BWA v0.7.12. MACS2 version 2.1.0 peak calling algorithm was used to identify regions of ChIP enrichment over background. An enrichment q-value threshold of <0.05 was used for all data sets. PeakAnnotator was used to identify the nearest transcription start site (TSS) of every peak.

## Tumor models and bioluminescence imaging

The institutional animal care and use committee of Ren Ji Hospital approved all animal protocols (permit-number: m20170205) and all animal experiments were in accordance with Ren Ji Hospital policies on the care, welfare, and treatment of laboratory animals. Six-week-old BALB/c Nude mice were used for in vivo studies. HEY or SKOV3 luciferase-labeled tumor cells ($1 \times 10^6$) were intraperitoneally injected to the nude mice. For treatment of tumor-bearing mice, panobinostat was given at a dose of 2.5 mg/kg/day and romidepsin was given at a dose of 1.5 mg/kg every other day. Tumor growth was monitored by bioluminescence imaging twice a week. For imaging, mice were injected with 1.5 mg of D-luciferin (15 mg/ml in PBS) and then anesthetized with isoflurane. Images were acquired within 8 min after injection with an IVIS Spectrum CT instrument coupled to Living Image acquisition software (PerkinElmer). Images were analyzed with Living Image software (version 4.5). Bioluminescent flux (photons/sec/cm$^2$/steradian) was determined for all mice in a prone position. All the PDX models were generated in NOD SCID mice using tumor tissues acquired during surgical resection, with prior written informed consent obtained from the patients. The initial diagnosis for both studied models (EOC-002 and EOC-004) was high-grade serous ovarian cancer, and the established PDX tumors were subcutaneously implanted in the dorsal flank of BALB/c Nude mice. When

tumor sizes reached 150–250 mm$^3$, animals were randomized into four groups of five mice each. One group of mice was treated with vehicle control (0.25% DMSO and 5% glucose), and the other three groups were treated with panobinostat (2 mg/kg/day), cisplatin (2 mg/kg/week) and a combination of both drugs, respectively. Tumor volumes were measured with a caliper and calculated as length $\times$ width$^2$ $\times$ 0.52.

## Statistical analysis

Gene set enrichment analysis was performed using the GSEA software (*Subramanian et al., 2005*). Gene ontology and pathway analyses were performed with DAVID Bioinformatics Resources (*Huang et al., 2009*). TCGA data were downloaded from cbioPortal or based on our previous analysis (*Yin et al., 2016*; *Zhang et al., 2015*). Statistical analysis was performed with GraphPad Prism software version 6.0. In all experiments, comparisons between two groups were based on two-sided Student's t-test and one-way analysis of variance (ANOVA) was used to test for differences among more groups. p-Values of <0.05 were considered statistically significant.

## Additional information

### Competing interests

Ying Yan, Zhenyu Gu: is an employee of GenenDesign Co. Ltd. The author declares no other competing interests exist. The other authors declare that no competing interests exist.

### Funding

| Funder | Grant reference number | Author |
| --- | --- | --- |
| National Natural Science Foundation of China | 81472537 | Guanglei Zhuang |
| National Natural Science Foundation of China | 81772770 | Wen Di |
| National Natural Science Foundation of China | 81802584 | Meiying Zhang |
| National Natural Science Foundation of China | 81802734 | Pengfei Ma |
| National Natural Science Foundation of China | 81802809 | Mei-Chun Cai |
| Shanghai Municipal Education Commission-Gaofeng Clinical Medicine Grant Support | 20161313 | Guanglei Zhuang |
| Shanghai Rising-Star Program | 16QA1403600 | Guanglei Zhuang |
| Shanghai Municipal Commission of Health and Family Planning | 2017ZZ02016,ZY(2018-2020)-FWTX-3006 | Wen Di |
| Science and Technology Commission of Shanghai Municipality | 16140904401 | Xia Yin |
| Shanghai Municipal Commission of Health and Family Planning | 20174Y0043 | Mei-Chun Cai |
| Program of Shanghai Hospital Development Center | 16CR2001A | Wen Di |
| School of Medicine, Shanghai Jiao Tong University | YG2016MS51 | Xia Yin |
| State Key Laboratory of Oncogenes and Related Genes | SB17-06 | Mei-Chun Cai |
| Shanghai Sailing Program | 18YF1413200 | Pengfei Ma |

| National Key R&D Program of China | 2016YFC1302900 | Wen Di |
| Science and Technology Commission of Shanghai Municipality | 18441904800 | Wen Di |
| The Shanghai Institutions of Higher Learning | Eastern Scholar | Guanglei Zhuang |
| National Natural Science Foundation of China | 81672714 | Guanglei Zhuang |

The funders had no role in study design, data collection and interpretation, or the decision to submit the work for publication.

### Author contributions

Kaixuan Shi, Resources, Data curation, Software, Formal analysis, Investigation; Xia Yin, Resources, Data curation, Funding acquisition; Mei-Chun Cai, Software, Formal analysis, Funding acquisition, Investigation; Ying Yan, Chenqiang Jia, Pengfei Ma, Shengzhe Zhang, Resources, Investigation; Zhenfeng Zhang, Zhenyu Gu, Resources; Meiying Zhang, Resources, Funding acquisition, Project administration; Wen Di, Conceptualization, Resources, Funding acquisition; Guanglei Zhuang, Conceptualization, Resources, Formal analysis, Supervision, Funding acquisition, Writing—original draft, Project administration

### Author ORCIDs

Guanglei Zhuang (iD) https://orcid.org/0000-0001-8141-5096

### Ethics

Animal experimentation: The institutional animal care and use committee of Ren Ji Hospital approved all animal protocols (permit-number: m20170205) and all animal experiments were in accordance with Ren Ji Hospital policies on the care, welfare, and treatment of laboratory animals.

### Decision letter and Author response

Decision letter https://doi.org/10.7554/eLife.44306.029
Author response https://doi.org/10.7554/eLife.44306.030

## Additional files

### Supplementary files

• Supplementary file 1. Cancer outlier genes.
DOI: https://doi.org/10.7554/eLife.44306.020

• Supplementary file 2. TMA patient information.
DOI: https://doi.org/10.7554/eLife.44306.021

• Supplementary file 3. Screening of 180 inhibitors in KURAMOCHI cells.
DOI: https://doi.org/10.7554/eLife.44306.022

• Supplementary file 4. PAX8 gene signature expression.
DOI: https://doi.org/10.7554/eLife.44306.023

• Supplementary file 5. PCR primers, sgRNA sequences and siRNA sequences.
DOI: https://doi.org/10.7554/eLife.44306.024

• Transparent reporting form
DOI: https://doi.org/10.7554/eLife.44306.025

### Data availability

The sequencing data have been deposited in NCBI SRA database (http://www.ncbi.nlm.nih.gov/sra/) under the accession number SRP153266.

The following dataset was generated:

| Author(s) | Year | Dataset title | Dataset URL | Database and Identifier |
|---|---|---|---|---|
| Shi K, Yin X, Cai MC, Yan Y | 2018 | RNAseq of ovarian cancer cell lines: HDAC inhibitors,sgPAX8 treatment | https://www.ncbi.nlm.nih.gov/sra/?term=SRP153266 | NCBI Sequence Read Archive, SRP153266 |

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
