## [Decision Letter]

Thank you for submitting your article "PAX8 regulon in ovarian cancer links lineage dependency with epigenetic vulnerability to HDAC inhibitors" for consideration by *eLife*. Your article has been reviewed by three peer reviewers, including Wilbert Zwart as the Reviewing Editor and Reviewer #1, and the evaluation has been overseen by Maarten van Lohuizen as the Senior Editor. The following individual involved in review of your submission has also agreed to reveal their identity: Ronny Drapkin (Reviewer #2).

The reviewers have discussed the reviews with one another and the Reviewing Editor has drafted this decision to help you prepare a revised submission.

Summary:

In this manuscript, the authors use a modified cancer outlier profile analysis (COPA) to identify the PAX8 ovarian lineage-specific regulon using RNA-Seq data from 700 human cancer cell lines. The analyses also identified 8 other lineage-restricted outliers in ovarian cancer, including FGF18. The authors show that PAX8 directly binds to the FGF18 promoter and regulates its expression and that the PAX8-FGF18 axis is responsible for promoting cell migration in an autocrine fashion. An image-based drug screen pinpointed that PAX8 expression was potently inhibited by small-molecule inhibitors of histone deacetylases (HDACs). in vitro and in vivo experiments showed that treatment with HDAC inhibitors led to altered histone H3K27 acetylation, perturbed the super-enhancer topology associated with the PAX8 gene locus, downregulated PAX8 transcription and its related targets, and led to decreased migration and tumor growth.

Essential revisions:

1) The effect of knocking out all three HDACs on PAX8 protein levels is not nearly as impressive as treatment with HDAC inhibitors. The effect is ~50% reduction in PAX8 levels. This should be discussed as it is not as corroborative as stated. Figure 6 shows synergy between cisplatin treatment and HDAC inhibition, and the authors connect this to PAX8. However, as HDAC inhibition has rather wide-spread impact, it is not yet convincingly presented that the effects observed in Figures 3 and 6 are truly PAX8-mediated. Could the phenotype be reverted by exogenous introduction of PAX8? Would expressing PAX8 cDNA from a different promoter (e.g. lentiviral LTR) bypass HDACi-mediated transcriptional suppression? If this mechanism is relevant, one would expect some degree of rescue in the growth/motility assays shown in Figure 3G-J.

2) The choice of cell line models needs to be better explained, and validation experiments to confirm reproducibility of findings between cell lines need to be provided. Figure 5: Why is Kuramochi not used in these animal studies since it was one of the main cell lines used in the in vitro studies? The effect of knocking out PAX8 in HEY cells is not as convincing as in SKOV3 cells or as in the HDACi-treated cell. In Figure 2, OCTOKO cells are used while the imaging screen for PAX8 levels was performed in KURAMOCHI cells. Validations experiments illustrating that key results from Figure 2 are also reproducible in KURAMOCHI cells should be included. "We first conducted a microarray experiment on OVTOKO cells transduced with PAX8 small interfering RNA (siRNA) or scrambled control (Figure 2A)". Why did the authors choose this cell line? OVTOKO harbors a PAX8 missense mutation at residue P350 that is unusual and is not representative in the population. Moreover, OVTOKO is considered a clear cell carcinoma cell line, not a high-grade serous carcinoma line, like KURAMOCHI, HEY and SKOV3.

3) What is the effect on cell viability and migration in the triple HDAC1-3 knockout cells? Furthermore, are the authors proposing that HDAC1-3 function as transcriptional activators of the PAX8 locus? Do these proteins occupy the enhancers at the PAX8 locus? This issue should be discussed and addressed experimentally. Otherwise, there are no real mechanistic concepts in this study to advance our understanding of HDACi-cancer phenomenology.

4) As effects of HDAC inhibitors are likely genome-wide, why would PAX8 be specifically affected? We would encourage the author to perform a time-course assay that measures mature mRNA and nascent RNA following HDACi exposure (including spike-in controls). The rationale would be to determine whether PAX8 is truly a hypersensitive gene to this drug perturbation. The heatmap shown in Figure 4—figure supplement 1C implies that many genes are sensitive this perturbation, and I think it is important to determine the ranking of PAX8 relative to all other expressed genes, with respect to its sensitivity to HDACi.

[Editors' note: further revisions were requested prior to acceptance, as described below.]

Thank you for resubmitting your work entitled "PAX8 regulon in human ovarian cancer links lineage dependency with epigenetic vulnerability to HDAC inhibitors" for further consideration at *eLife*. Your revised article has been favorably evaluated by Maarten van Lohuizen as the Senior Editor, and Wilbert Zwart as the Reviewing Editor.

The manuscript has been improved but there are some remaining issues that need to be addressed before acceptance, as outlined below:

In the rebuttal letter, referencing Figure 4—figure supplement 1, it is stated that 'Indeed, H3K27ac occupancy profiles unveiled a typical super-enhancer region around PAX8 promoter (Figure 4A), which was perturbed following HDAC inhibition as indicated by disproportional H3K27ac enrichment within PAX8 introns (Figure 4B; revised Figure 4—figure supplement 1A)'.

However, as no y-axis is provided in this figure, it cannot be stated whether the overall signal is indeed increased, giving rise to the disproportional H3K27ac enrichment at PAX8 introns (as stated by the authors), or whether the overall signal is actually going down and the peaks at the PAX8 promoter are lost. Please provide tag count information in the figure y-axis, so that the conclusion is better supported by the data visualised.

---

## [Author Response]

Essential revisions:1) The effect of knocking out all three HDACs on PAX8 protein levels is not nearly as impressive as treatment with HDAC inhibitors. The effect is ~50% reduction in PAX8 levels. This should be discussed as it is not as corroborative as stated. Figure 6 shows synergy between cisplatin treatment and HDAC inhibition, and the authors connect this to PAX8. However, as HDAC inhibition has rather wide-spread impact, it is not yet convincingly presented that the effects observed in Figures 3 and 6 are truly PAX8-mediated. Could the phenotype be reverted by exogenous introduction of PAX8? Would expressing PAX8 cDNA from a different promoter (e.g. lentiviral LTR) bypass HDACi-mediated transcriptional suppression? If this mechanism is relevant, one would expect some degree of rescue in the growth/motility assays shown in Figure 3G-J.

We agree with the reviewers that the effect of knocking out HDAC1-3 on PAX8 levels is less striking than HDAC inhibitors. At least two reasons may lead to these reproducible results in two different ovarian cancer cell lines. First, HDAC1-3 depletion was not complete, as evidenced by low amount of detectable proteins in Western blots (Figure 3—figure supplement 2A). CRISPR/Cas9-mediated gene editing is known to contain in-frame repairs and cannot achieve 100% efficiency, especially when accompanied by impaired cell viability due to HDAC knockout (revised Figure 3—figure supplement 2B). Second, PAX8 downregulation was anti-proliferative and might be selected against through alternative PAX8 activating machinery. The same selection was seemingly negligible with HDAC inhibitors, conceivably due to their rapid mode of action relative to the genetic approach.

To determine whether the observed effects of HDAC inhibition were genuinely mediated by PAX8, we conducted the rescue studies by overexpressing PAX8 via lentiviral transduction in KURAMOCHI, HEY, SKOV3 and OVTOKO cells. Two critical findings were noted in these experiments. First, exogenous PAX8 expression using the lentiviral vector was no longer antagonized by HDAC inhibitors, consistent with our proposed model of epigenetic regulation on PAX8 expression by HDAC molecules. Of particular interest, panobinostat or romidepsin treatment counterintuitively resulted in markedly elevated PAX8 levels in this setting (revised Figure 3—figure supplement 2C), reminiscent of the recently reported interplay between HDAC and YAP (Han H., et al., Oncogene. 2018.). Second, the inhibitory effects of HDAC compounds on cell viability and motility were partially reversed following PAX8 re-expression (revised Figure 3—figure supplement 2D). Taken together, our data supported a role of HDAC proteins in regulating PAX8 expression and function.

2) The choice of cell line models needs to be better explained, and validation experiments to confirm reproducibility of findings between cell lines need to be provided. Figure 5: Why is Kuramochi not used in these animal studies since it was one of the main cell lines used in the in vitro studies? The effect of knocking out PAX8 in HEY cells is not as convincing as in SKOV3 cells or as in the HDACi-treated cell. In Figure 2, OCTOKO cells are used while the imaging screen for PAX8 levels was performed in KURAMOCHI cells. Validations experiments illustrating that key results from Figure 2 are also reproducible in KURAMOCHI cells should be included. "We first conducted a microarray experiment on OVTOKO cells transduced with PAX8 small interfering RNA (siRNA) or scrambled control (Figure 2A)". Why did the authors choose this cell line? OVTOKO harbors a PAX8 missense mutation at residue P350 that is unusual and is not representative in the population. Moreover, OVTOKO is considered a clear cell carcinoma cell line, not a high-grade serous carcinoma line, like KURAMOCHI, HEY and SKOV3.

We thank the reviewers for the insightful comments. In most cases, multiple cell lines (often KURAMOCHI, HEY, SKOV3 and OVTOKO) were used to represent histopathologically and genetically heterogeneous ovarian cancer. A number of factors contributed to the specific choice of cell lines occasionally. For example, OVTOKO exhibited dependency on PAX8 to a relatively larger extent than others. Therefore, it was chosen to perform the first sets of experiments including microarray profiling, regulon identification, and phenotypic assay of regulon genes, in order to yield more reliable findings. On the other hand, KURAMOCHI grew better in culture and had a unique feature of giant round nuclear morphology, which suited well for the image-based high-throughput screen. We have also tested the ovarian cell lines in BALB/c Nude mice, and only HEY and SKOV3 efficiently formed xenografts. Because PAX8 was identified as a lineage-survival oncogene in epithelial ovarian cancer of Müllerian tract origin, both prevailing high-grade serous carcinoma and scarce clear cell carcinoma models were appropriate for the study. We found that the PAX8 missense mutation at residue P350 in OVTOKO did not appear to be associated with functional deficiency of PAX8 or altered regulation by HDACs, and thus would unlikely affect our major conclusions, all of which were further verified by independent cells lines with wildtype PAX8.

To improve our manuscript, we have now applied several approaches to underline the experimental reproducibility. First, the phenotypic screen of regulon genes in OVTOKO cells (Figure 2F) was repeated in KURAMOCHI cells (revised Figure 2F). Second, the diminished PAX8 immunofluorescent signal in panobinostat-treated KURAMOCHI was also observed in HEY, SKOV3 and OVTOKO (revised Figure 3—figure supplement 1B). Third, pharmacologic HDACs inhibition was conducted with two additional compounds (revised Figure 3—figure supplement 1C). Forth, we provided scientific rationale for the choice of cell lines wherever applicable.

3) What is the effect on cell viability and migration in the triple HDAC1-3 knockout cells? Furthermore, are the authors proposing that HDAC1-3 function as transcriptional activators of the PAX8 locus? Do these proteins occupy the enhancers at the PAX8 locus? This issue should be discussed and addressed experimentally. Otherwise, there are no real mechanistic concepts in this study to advance our understanding of HDACi-cancer phenomenology.

This is an important point, and we appreciate the reviewers’ questions. We conducted cell proliferation and migration assays, and confirmed that the triple HDAC1-3 knockout cells exhibited impaired viability and defective motility in comparison to control cells (revised Figure 3—figure supplement 2B), consistent with the observed PAX8 downregulation upon HDAC1-3 depletion. Mechanistically, considering their canonical histone deacetylase activity, we reasoned that class I HDAC proteins would less likely specifically bind to the PAX8 gene locus and function as transcriptional activators, but rather indirectly regulate PAX8 expression by epigenetically controlling the H3K27ac-marked enhancer topology. Indeed, H3K27ac occupancy profiles unveiled a typical super-enhancer region around PAX8 promoter (Figure 4A), which was perturbed following HDAC inhibition as indicated by disproportional H3K27ac enrichment within PAX8 introns (Figure 4B; revised Figure 4—figure supplement 1A). Since super-enhancer was characteristic of many lineage-survival oncogenes, they might be uniformly susceptible to epigenetic interventions. Our preliminary tests on NKX2-1 in lung adenocarcinoma and *SOX2* in lung squamous carcinoma supported this hypothesis (Figure 4—figure supplement 2). Therefore, the mechanistic concept in our study shed new light on the lineage-dependency model across human cancers, and held the promise to target lineage-survival oncogenes in often difficult-to-treat neoplasms. We have added the new data and revised the manuscript to highlight the above point.

4) As effects of HDAC inhibitors are likely genome-wide, why would PAX8 be specifically affected? We would encourage the author to perform a time-course assay that measures mature mRNA and nascent RNA following HDACi exposure (including spike-in controls). The rationale would be to determine whether PAX8 is truly a hypersensitive gene to this drug perturbation. The heatmap shown in Figure 4—figure supplement 1C implies that many genes are sensitive this perturbation, and I think it is important to determine the ranking of PAX8 relative to all other expressed genes, with respect to its sensitivity to HDACi.

These are excellent suggestions. By means of RNAseq analysis, we identified 7693 (4442 up and 3251 down) and 7764 (4470 up and 3294 down) differentially expressed genes (fold change > 2; adjusted p-value < 0.05) related to panobinostat and romidepsin treatment, respectively (Figure 4—figure supplement 1C). Therefore, the effects of HDAC inhibitors were indeed genome-wide in ovarian cancer cells. To determine the ranking of PAX8 relative to other expressed genes, we accepted the reviewer’s advice to perform a time-course assay of total mRNA or nascent RNA following HDACi exposure. Unfortunately, the experiment on nascent RNA was not successful due to technical challenges and we are still trying to solve the problem. Meantime, the time-course assay of total mRNA unambiguously demonstrated that PAX8 was among the hypersensitive genes to HDAC inhibitors (revised Figure 4—figure supplement 1B). We understood that these results could be confounded by RNA stability, and thus used the general transcription inhibitor actinomycin D as a negative control. These data revealed that HDAC inhibition, to some degree, specifically affected PAX8 gene expression.

[Editors' note: further revisions were requested prior to acceptance, as described below.]

The manuscript has been improved but there are some remaining issues that need to be addressed before acceptance, as outlined below:In the rebuttal letter, referencing Figure 4—figure supplement 1, it is stated that 'Indeed, H3K27ac occupancy profiles unveiled a typical super-enhancer region around PAX8 promoter (Figure 4A), which was perturbed following HDAC inhibition as indicated by disproportional H3K27ac enrichment within PAX8 introns (Figure 4B; revised Figure 4—figure supplement 1A)'.However, as no y-axis is provided in this figure, it cannot be stated whether the overall signal is indeed increased, giving rise to the disproportional H3K27ac enrichment at PAX8 introns (as stated by the authors), or whether the overall signal is actually going down and the peaks at the PAX8 promoter are lost. Please provide tag count information in the figure y-axis, so that the conclusion is better supported by the data visualised.

We thank the reviewers for pointing this out, and have reset the automatic y-axis to a fixed scale for direct comparison of three conditions. Interestingly, the initially observed relative H3K27ac enrichment within PAX8 introns upon HDAC inhibition is indeed driven by the loss of ChIP-seq peaks around the gene promoter. This phenomenon turns out to be widespread as shown in Author response image 1, which presents the H3K27ac marks adjacent to the transcription start sites (TSS).

Therefore, HDAC inhibitors might lead to a re-distribution of histone acetylation, at the expense of appropriate localization. Although these results are unexpected given HDACs’ best known function as acyl-lysine “erasers”, a possible mechanistic explanation is that the histone acetylation-deacetylation circuitry could be simultaneously disrupted, such that H3K27ac signals become generally flat throughout the chromatin and the enhancer topology is consequently perturbed. We have revised the description to better reflect the ChIP-seq data.